# Hydrogeochemical Characterization and Identification of Factors Influencing Groundwater Quality in Coastal Aquifers, Case: La Yarada, Tacna, Peru

**DOI:** 10.3390/ijerph19052815

**Published:** 2022-02-28

**Authors:** Samuel Chucuya, Alissa Vera, Edwin Pino-Vargas, André Steenken, Jürgen Mahlknecht, Isaac Montalván

**Affiliations:** 1Department of Geology-Geotechnics, Jorge Basadre Grohmann National University, Tacna 23000, Peru; schucuyam@unjbg.edu.pe; 2Department of Civil Engineering, Jorge Basadre Grohmann National University, Tacna 23000, Peru; averam@unjbg.edu.pe (A.V.); imontalvand@unjbg.edu.pe (I.M.); 3School of Engineering and Sciences, Tecnologico de Monterrey, Monterrey 64849, Nuevo Leon, Mexico; jurgen@tec.mx

**Keywords:** coastal aquifer, seawater intrusion, salinization, hydrochemical signatures

## Abstract

The coastal aquifer La Yarada has anthropogenic and geogenic contamination that adversely affect the quality of groundwater for population and agricultural use. In this scenario, multivariate statistical methods were applied in 20 physicochemical and isotopic parameters of 53 groundwater pumping wells in October 2020, with the aim of characterizing the hydrogeochemical processes that dominate the groundwater of the coastal aquifer and the factors that cause them to optimize the effective management of water resources, delimiting areas affected by more than one salinization process. The samples were grouped into three clusters (C1, C2, and C3) with cluster analysis, the spatial distribution of C2 and C3 (reclassified in stiff diagrams), evidenced hydrogeochemical facies associated with the flow and recharge directions governed by the structural lineaments (NE-SO), favoring some areas more than others, arising different facies and hydrogeochemical processes. Factor analysis was applied from three different approaches: (1) main elements, (2) trace elements, and (3) physicochemical and isotopic parameters; exposing 6 distinguishable hydrogeochemical processes in the aquifer and factors that cause them: (i) salinization—marine intrusion, (ii) fertilizer leaching and dissolution of (Ca^2+^, Mg^2+^), (iii) wastewater mixture (NO^3−^), (iv) reducing conditions (Fe, Mn, Al), (v) contributions of (B, Sr), (vi) conservative mixtures and dissolution (As, F). It was validated with water quality indices (WQI) according to the national limits, delimiting 67 km^2^ parallel to the coast with “bad” to “very bad” quality for human consumption and unsuitable for irrigation according to the Wilcox diagram thus pre-treatment in this area is indispensable.

## 1. Introduction

Marine intrusion and salinization of groundwater are common problems that affect many arid and semi-arid coastal areas, such as the aquifer of La Yarada in the Tacna region [1]. In the absence of surface water resources, farmers in the flatlands of La Yarada use groundwater as their only source of irrigation, reaching levels of overexploitation of the aquifer [2,3,4]. This leads to a gradual decrease in the water table and, as a consequence, the quality of the water is affected by marine intrusion and by reducing secondary recharges of good quality, being confined to receiving the main recharge from the Caplina River, that presents neogenic volcanic activity in the source of its basin [5,6]. This deterioration of water quality seriously affects agricultural productivity, the aquatic environment, and human health [7].

The hydrochemical signatures of the groundwater are the result of hydrogeochemical processes and the spatio-temporal distribution, the rainwater that recharges the aquifer has low ionic content in the surface runoff, but in the zone saturated by water–rock interaction acquires chemical substances as dissolved species [6,8], anthropogenic activities, and the freshwater-seawater interaction existing in semi-arid coastal aquifers and also change the hydrochemical composition of groundwater [9,10].

Multivariate statistical analysis (cluster analysis and factor analysis) together with ionic relations analysis represent solid tools for detecting and explaining the hydrogeochemical processes that govern the hydrochemical composition of groundwater in the aquifer [3,9,10,11]. As a result of multivariate analysis applying chemical and isotopic variables [3], two processes were identified mainly of seawater front movement: dispersion (diffusion) of chemical elements and different types of water mixing. Defining and delimiting the influence of each hydrogeochemical process in La Yarada aquifer by an analysis from three different approaches: (1) main elements, (2) trace elements, and (3) physicochemical and isotopic parameters, is, therefore, an important step for the identification of the “factors that influence water quality.” Management requires the collection and analysis of large water quality data sets that can be difficult to assess and synthesize. A water quality index (WQI) is an efficient tool to identify the quality of water and its suitability for a particular use [12,13,14]. WQI models are based on aggregation functions that allow the analysis of large temporally and spatially varying water quality datasets to produce a single value [15].

## 2. Study Area

The coastal aquifer La Yarada located in the headwaters of the Atacama Desert, is located in southern Peru and partially in the north of Chile and comprises 750 km^2^ of the lower and middle basin of the Caplina River [16] (Figure 1).

### 2.1. Climate and Vegetation

The region is dominated by a desert climate, with moderate temperature changes [1] locally. It presents scarce pluvial precipitations in the form of drizzle produced by dense fogs rising from the coast towards the flatlands of La Yarada and Hospicio. Regionally, the maximum rainfall recorded in the stations of Palca and Toquepala during the last 20 years has reached 129 mm per year, experiencing water scarcity compared to the average monthly flow of the rivers of the Peruvian coast, which have much higher average flows [17,18].

The cultivated lands in the aquifer of La Yarada correspond to the Caplina Valley, partially irrigated by the Caplina River and the area of flatlands of La Yarada and Hospice with groundwater irrigation pumped own aquifer (Figure 2) [17]. The Municipal Forest represents an extensive forest massif in the area of 60 Ha, of artificial origin created through the sustainable use of wastewater treated in the Magollo treatment plant, located in the border area of Magollo and La Yarada [19,20].

### 2.2. Hydrogeological Settings

The geology (Figure 3a) and geomorphology (Figure 3b) of the site result in a complex groundwater system in the Caplina Basin. The aquifer of La Yarada of alluvial and fluvial origin of Quaternary age, lithologically formed by gravel, fine to coarse sands, silt and clays (Figure 3a) intercalated set up a multi-layered aquifer (Figure 3c) made up of unconsolidated fluvial strata interspersed with stratifications of the Huaylillas volcanic impermeable formation (Nm-hu) and below it the stratifications of the Moquegua formation (Po-mo) of detrital material (Figure 3c) [22].

Structural controls govern underground flow directions NE-SO in the area involving the aquifer of La Yarada (Figure 3a) [18] with hydraulic gradients in order of 10^−2^ and with depth to the water table of 50 m in its middle zone [22]. Local geophysical studies allow postulating the existence of normal high-angle faults that produce a vertical displacement (≥200 m) between the outcropping stratigraphic sequences [23,24].

### 2.3. Hydrology

The average annual relative humidity in the flatlands of La Yarada and the city of Tacna is 75%, with monthly highs reaching 90% in the winter months and a monthly low reaching 55% in the summer months [20]. The maximum precipitation in the pampas of La Yarada and Hospicio barely reaches 2 mm in the months of June and July. The maximum temperatures reach 22.5 °C in the month of February [14].

The recharge of the aquifer is mainly due to precipitation in the humid zone adjacent to the Barroso Mountain range (Figure 4b), whose runoff passes through narrow ravines and expands in the fluvial range of the Yarada, the sub-basins that contribute to the recharge of the aquifer are Caplina, Palca, Uchusuma, Cobañi, Viñani, Cauñani, Espíritus and Escritos (Figure 4a). Another source of recharge is inputs generated by irrigation infiltration and wastewater irrigation in the Copare, Arunta, and Magollo sectors [22].

### 2.4. Water Balance

Irrigation in the flatlands of La Yarada began in 1954 (Figure 2). Since then, the areas irrigated with groundwater have increased exponentially (Figure 5a), along with the volumes of exploitation (Figure 5b). The overall groundwater level decline for the period 2002–2020 was 6.62 m [21]. Since 1967, there has been an extraction of 12.9 Hm^3^ with a favorable recharge balance; from 1971 the water balance was negative (Figure 5c). The values of overexploitation were increasing alarmingly, and for 2009 a recharge of 53 Hm^3^ was calculated, but the exploitation reached 97.5 Hm^3^ año^−1^, with a deficit of 44.5 Hm^3^. The panorama became more critical for 2019, reaching 197.1 Hm^3^ with the same recharge levels [21,25].

## 3. Materials and Methods

### 3.1. Monitoring Network and Sampling

The hydrochemical and isotopic monitoring network, distributed in the aquifer of La Yarada in October 2020, comprised 53 samples collected at the outlet level of the pumping wells. Electrical conductivity (EC) and pH parameters were measured in situ. Analysis of cations and total metals (Ca^2+^, Mg^2+^, Na^+^, K^+^, B, Mn, Fe, As, Sr, Al) was carried out by inductively coupled plasma mass spectrometry (ICP-MS) following ISO 17 294-2 method. The anions (SO_4_^2−^, Cl^−^, NO_3_^−^, F^−^) were analyzed by ion chromatography using EPA 300.0 method. The bicarbonate (HCO_3_^−^) was determined by the acid-base assay method. The δ^18^O and δ^2^H isotopes of the water samples were measured with the ABB-LGR analyzer Model 912-0008 with analytical accuracy of ±0.1 and ±1.0‰, respectively.

### 3.2. Data Quality

#### 3.2.1. Quality Analysis of Hydrochemical Data

For quality control and analytical accuracy, the concentrations of total cations and total anions of each sample were recalculated from (mg/L) to (mEq/L), the ionic equilibrium error was calculated using Equation (1), considering those samples that were within the acceptable limit of ±10% [26].
(1)$$\mathrm{Error}\;\left(\%\right)=\frac{\sum^{\;}\mathrm{cationes}-\sum^{\;}\mathrm{aniones}}{\sum^{\;}\mathrm{cationes}+\sum^{\;}\mathrm{aniones}}\times 100$$

#### 3.2.2. Statistical Preparation

The Anderson–Darling test was performed to identify those parameters that follow a normal Gaussian distribution. Data normalization was performed for parameters that do not have a normal distribution, a prerequisite for multivariate statistical analysis, to increase confidence in results and interpretations [27,28]. For the main elements together with the trace elements, the transformation of the natural logarithm data was applied [29]. However, for isotopes, three-parameter log normal was applied, ideal for negative numbers. In order to guarantee normalization, the Anderson–Darling test was reapplied.

Then with data being standardized, thus that each variable had the same weight in the analysis, this standardization process assembles the data as values without units to avoid the effects of the value dimensions [30,31]. All the procedures and statistical analyses were performed using the Python programming language and its various integrated packages.

### 3.3. Multivariate Statistical Analysis

#### 3.3.1. Cluster Analysis

Cluster analysis was applied to the physical parameters (ph, EC, and TDS), main ions (Cl^−^, SO_4_^2−^, NO_3_^−^, HCO_3_^−^, Ca^2+^, Mg^2+^, Na^+^, K^+^) and trace elements (B, Mn, Fe, As, Sr, F, and Al), using the Ward method with Euclidean distance, obtaining a dendrogram (tree diagram) that groups the groundwater wells according to their similarity in clusters. Graphs of facies and ionic relations (stiff diagram, comparative trace element diagram, Chadha diagram, and Simpson’s inverse salinization degrees) were spatially represented, identifying the differences and similarities in each group. These charts were most helpful when applied to previously aggregated data [32].

#### 3.3.2. Factor Analysis

Factor analysis was applied, using Kaiser’s criterion [33] in 3 different analysis approaches: (i) main elements (FP), (ii) trace elements (FO), (iii) physicochemical and isotopic parameters (FT), varimax rotation was performed to reduce variables with lower contribution, which will distinguish the physicochemical variables according to their degree of covariation, reducing the large number of variables to a smaller number of factors, represented by parameters with their respective factor load, being “strong” > ± 0.75, “moderate” for ±0.75–0.5 and weak for ±0.5–0.3 [34,35], these indicate the degree of contribution of the variables in their factors [36], and also factor scores indicating the intensity of the influence of each factor on the location of each sample. Factor scores were interpolated using Ordinary Kriging (OK), since it is the most common and robust geostatistical technique of unbiased estimator, recommended in hydrogeological studies for its simplicity and precision [37,38,39], for the application of this technique, the software QGIS 3.22, Geographic Information System, Open-Source Geospatial Foundation Project was used, with the best semi-variogram models (exponential, Gaussian, and spherical), helping to build accurate prediction maps.

### 3.4. Water Quality

The quality of water for drinking use will be assessed using the water quality index (WQI) method [40], which combines the effects of the parameters in comparison with the standard limits prescribed by the Ministry of Health of Peru [41]. The calculation of WQI consists of 4 steps: (1) assigning a weight to the parameters based on their relative effects on water quality, (2) calculation of the relative weight of each parameter as the relationship between the assigned weight and the total weight (Equation (2)), (3) calculation of a quality rating scale for each parameter in each water sample (Equation (3)), (4) multiplication of the calculated relative weight and the quality rating scale, to find the water quality sub-index (Equation (4)), (5) sum of all water quality sub-indices for each water quality parameter, resulting in the WQI of the water sample (Equation (5)). Once the indices were obtained, their spatial distribution was represented and classified as excellent, good, bad, very bad, and inadequate [42].
(2)Wi=wi∑i=1nwi
(3)$$q_{i}=\left(\frac{C_{i}}{S_{i}}\right)\times 100$$
(4)$${\mathrm{SI}}_{i}={\;W}_{i}\times {\;q}_{i}$$
(5)$$\mathrm{WQI}=\overset{n}{\underset{i=1}{\sum }}{\mathrm{SI}}_{i}$$
where i stands for water quality parameter, W_i_ for relative weight, w_i_ for assigned weight, n for the number of parameters, q_i_ for quality rating scale, C_i_ for measured concentration, SI for MINSA prescribed drinking limit, and SI_i_ for water quality sub-index. For the calculation of WQI, the concentrations of water quality parameters in mg/L were utilized.

To evaluate water quality for irrigation purposes, the Wilcox diagram was used [43], which uses percent sodium (Na%) obtained by Equation (6), classifying the water into 5 categories (excellent, good, acceptable, ordinary, and inadequate). This graph was applied with the data previously grouped by the cluster analysis, identifying the differences and similarities of these.
(6)$$\mathrm{Na}\;\%=\frac{\left({\mathrm{Na}}^{+}+K^{+}\right)}{\left({\mathrm{Na}}^{+}+K^{+}+{\mathrm{Ca}}^{2+}+{\mathrm{Mg}}^{2+}\right)}\times 100$$

## 4. Results and Discussion

### 4.1. Hydrogeochemical Grouping

The 53 groundwater samples were grouped according to their similarity using cluster analysis. In Figure 6, the results are shown in the dendrogram (tree diagram). They were divided into three branches (groups) with a bond distance of 15. Each group represents a face that reflects the effect of a hydrogeochemical process or a combination of processes on the samples belonging to the group. Groups C2 and C3 have a shorter bond distance. Therefore, they are groups with less dissimilarity compared to group C1, with a high bond distance, revealing that they differ much more. These wide variations in bond distance between groups confirm variation in hydrogeochemical processes in the study area.

Figure 7a shows the spatial distribution of the three reclassified clusters with Stiff diagrams. Figure 7b organizes a comparative trace element diagram, allowing the differences and similarities between each group to be identified.

Cluster 1 is characterized by having the lowest concentration of ions according to the stiff diagram (Figure 7a), indicating that it is low mineralized water with short residence times. Cluster 3 with a similar shape in the stiff diagram but is much more polarized towards Cl^−^ and Ca^2+^ mineralizations, mostly influenced by marine intrusion due to its proximity to the coast and contamination by wastewater leakage in the area of the treatment plant. Cluster 2 wells, located on the boundaries of the aquifer, also present high concentrations of Cl^−^, associated with mixing with seawater. The area covered by this cluster is among the least recharged in the aquifer, while the samples from cluster 3 are relatively high because they receive the greatest recharge from the aquifer from the Caplina River [44], which has a preferential direction dominated by structural lineaments in the aquifer of La Yarada (Figure 7a) [18,23].

The Caplina river transports considerable concentrations of trace elements to the aquifer, originating from volcanic activity at the source of its basin [44], reflecting high concentrations of B in the three clusters (>1.5 mg/L) (Figure 7b) and because it is an element with high geochemical mobility. The wells located in cluster 3 contain a higher concentration of B and Sr than the other clusters, which reflects marine intrusion and a long water flow path [45]. Cluster 1 has the highest concentrations of Fe, Mn, and Al (Figure 7b), the result of reducing conditions due to hydrothermal coming from the Caplina River, which in its long flow through the subsoil depletes them of oxygen [46]. Cluster 2 stands out for the high concentrations of As and F, their location at the edges of the aquifer, the relative low recharge and distance from the structural lineament and as there are low concentrations in the main recharge zone from the Caplina River, suggest a local meteoric origin in groundwater due to dissolution of feldspars and biotite on contact with rhyolitic-ignimbrites and solutions from seawater intrusion with long residence times [47].

#### Ionic Relations

Groundwater samples were classified according to the relative percentage of their chemical components using the Chadha diagram (Figure 8a). A total of 80% of the samples were located in the Ca-Cl facie (80%), the remaining in the Na-Cl facie predominantly the wells of the C2 cluster and partially of the C1 in Na-Cl, revealing that the groundwater of the aquifer of La Yarada was mainly saline, this can be attributed to the presence of multiple sources of salinization that impact the chemistry of the groundwater and processes of direct and reverse ion exchange between Na and Ca, the product of the interaction between water and the medium.

The HCO_3_^−^/Cl^−^ vs. Cl^−^ ratio was used to analyze the degree of salinization (Figure 8b), based on the Simpson inverse [48]. Salinization in the aquifer ranges from moderate to severe, with the most acceptable levels being the samples included in cluster C1 and the most harmful in cluster C3, due to the influence of multiple sources of salinization and contamination that impact groundwater.

### 4.2. Hydrochemical Associations

In order to examine the relationship between variables in the samples, the factor analysis method was used, reducing factors made up of associations of variables. Using Kaiser’s criterion [33], it was applied in three different analysis approaches (i) main elements (FP), (ii) trace elements (FO), (iii) physicochemical and isotopic parameters (FT), processed in Table 1, Table 2 and Table 3, respectively.

#### 4.2.1. Main Elements (FP)

The main ions Cl^−^, SO_4_^2−^, NO_3_^−^, HCO_3_^−^, Ca^2+^, Mg^2+^, Na^+^, K^+^, as well as pH, EC, and TDS were considered. Three factors were obtained with a confidence interval of 95% and an explained total variance equal to 83% (Table 1).

The FP-1 factor explains 33% of the total variance and presents a positive correlation between EC, TDS, Cl^−^, Na^+^, and K^+^. Figure 9a shows its spatial distribution parallel to the sea line, the location of high charges on these variables represents the main ions in seawater, which would reflect the influence of marine intrusion on groundwater [2,49]. The FP-2 factor explains 32% of the total variance presents a positive correlation between Ca^2+^, Mg^2+^, SO_4_^2−^ and negative pH. Figure 9b shows its spatial distribution. This ion association can be attributed to the fact that irrigation favors the leaching of SO_4_^2−^ from agricultural fertilizers (CaSO_4_) [50].

The infiltrated water interacts with the soil matrix, raising salinity levels, decreasing the pH in the subsoil, generating dissolution of carbonate minerals present as cementing material in the alluvial deposits that make up the aquifer, enriching the groundwater with Ca^2+^ and Mg^2+^ [51,52]. The enrichment of Ca^2+^ and Mg^2+^ ions in areas susceptible to marine intrusion is due to the dominance of the reverse ion exchange process, which led to an excess of Ca^2+^ and a deficiency of Na^+^ [53]. The FP-3 factor explains 18% of the total variance, with a positive correlation between NO_3_^−^ and HCO_3_^−^ and a negative pH. Figure 9c shows its spatial distribution. The high charge of NO_3_^−^ in this factor, evidence nitrification product of microbial oxidation of NH_4_^+^ in the study area due to wastewater input. This flow leads to acidity [49], decreasing the pH, causing dissolution of carbonate minerals, and enriching the water with HCO_3_^−^. However, another source of bicarbonate would come from the oxidation process of decaying organic matter and the respiration of roots [54]. 

#### 4.2.2. Trace Elements (FO)

The trace elements B, Mn, Fe, As, Sr, F, and Al, were considered. Three factors were obtained, with a confidence interval of 95% and with a total explained variance equal to 75% (Table 2).

The FO-1 factor explains 33% of the total variance and presents a positive correlation between Mn, Fe, and Al. Figure 10a shows its spatial distribution, this association of ions is influenced by reducing conditions and the means through which the flow circulates [46], evidenced contributions of hydrothermal origin from the recharge of the Caplina River, due to the long run of water flow through the subsoil, becoming depleted in oxygen, creating the ideal conditions for the mobility of these elements. Adsorption processes of (Mn, Fe, and Al) and phyllosilicates (clay) eliminate these chemical species in the solution, regulating the distribution and mobility of trace elements in aquifers [55,56]. The factor FO-2 explains 23% of the total variance shows a positive correlation between B and Sr. Figure 10b shows its spatial distribution. The concentrations of B in groundwater are related to hydrothermal contributions from the recharge of the Caplina River and the invasion of seawater in the coastal zone, exceed the national quality standards in 80% of the aquifer because it is an element with high geochemical mobility, high aqueous solubility, natural abundance and lack of effects to oxidation-reduction reactions. However, its concentration was decreased by adsorption to soils by clay minerals and alkaline pH, alleviating the high concentration but not sufficiently [57]. The main source is hydrothermalism in the recharge zone of the Caplina basin and the invasion of seawater in the coastal zone. The concentration of Sr reflects the marine intrusion and the long run of the water flow [45] because the saltwater of the sea penetrates the freshwater, calcium is released together with strontium [58,59]. The FO-3 factor explains 19% of the total variance shows a positive correlation between As and F. Figure 10c shows its spatial distribution. The recharges that infiltrate the Caplina river is the main source of trace elements to the aquifer [44], but it does not present considerable concentrations of As, this suggests elements with sorption capacity, which retain As, restricting its mobility and regulating its distribution in the aquifer [60]. The origin of arsenic is highly controversial. Is groundwater available in relatively high concentrations but without a recharge zone, reflecting that it comes from a local meteoric origin in groundwater [61]. As and F in groundwater is also related to the dissolution of feldspars and biotite, due to contact with rhyolitic-ignimbrites and solutions from seawater intrusion with long residence periods [47], oxidation conditions, low circulation velocity in that area, the preferable pH, favor this process [62,63].

#### 4.2.3. Physicochemical and Isotopic Parameters (FT)

The main ions, trace elements, physical properties, and stable isotopes were considered. Four factors were obtained, with a 95% confidence interval and a total explained variance equal to 78%. The results, together with the identification of factors, are shown in Table 3. Most of these variables correspond to the multivariate analysis of the study (3), which identified the same processes in FT 1 and FT 2 explained below.

The first factor FT-1 explains 33% of the total variance and presents a positive correlation between EC, TDS, Cl^−^, SO_4_^2−^, Ca^2+^, Mg^2+^, Na^+^, K^+^, B, and Sr. Figure 11a shows its spatial distribution, this factor groups the largest number of elements, these have a high correlation with TDS and EC, being responsible for the salinity of the groundwater in the aquifer. The FT-2 factor explains 17% of the total variance presents a positive correlation between δ^2^H, δ^18^O, As, F, and a low correlation with pH, Figure 11b shows its spatial distribution, in a similar way to the FO-3 factor, the isotopes δ^2^H and δ^18^O are included, which show regions of conservative mixtures of fresh water and seawater, due to the over-exploitation of wells in the area, where it exists low flow and recharge, given its distance from the structural lineaments that dominate them (Figure 7a). The FT-3 factor explains 15% of the total variance, it has a positive correlation between NO_3_^−^, HCO_3_^−^, Cl^−^ and Ca^2+^ and pH negative. Figure 11c shows its spatial distribution in a similar way to the FP-3 factor. Cl^−^ and Ca^2+^ ions are included in this factor, the result of increased salinity due to the effect of wastewater and dissolution of carbonate minerals in the cementing matrix of the soil. The fourth factor FT-4 explains 13% of the total variance, it has a positive correlation between Mn, Fe, and Al, and Figure 11d shows its spatial distribution. Similar to the FO-1 factor, it represents reducing conditions in groundwater.

### 4.3. Hydrogeochemical Processes

The contrast of the three factorial analysis approaches together with the cluster analysis, has revealed six hydrogeochemical processes in the aquifer and factors that cause them (Figure 12): (i) salinization—marine intrusion, (ii) fertilizer leaching and dissolution (Ca^2+^, Mg^2+^), (iii) wastewater mixture (NO_3_^−^), (iv) reducing conditions (Fe, Mn, Al), (v) contributions of (B, Sr), (vi) conservative mixtures and dissolution (As, F).

#### 4.3.1. Salinization by Marine Intrusion

Seawater intrusion is described by the factor FP-1 (Figure 9a), covering the littoral area and the edges of the aquifer (NW and SE ends), where they invade more internal areas. This suggests two processes of motion of the seafront and, therefore, processes of ion exchange. This is explained by the fact that at the edges, the recharge is lower, allowing greater mixing of seawater and freshwater. Unlike the central area of the aquifer dominated by geological structural lineaments, it receives greater recharge mitigating the mixing of seawater and freshwater. However, chemical species migrate from seawater to groundwater by diffusion processes and reverse ion exchange (Na^+^ by Ca^2+^) (Figure 7a). 

#### 4.3.2. Fertilizer Leaching and Dissolution (Ca^2+^, Mg^2+^)

The leaching of agricultural fertilizers and dissolution of Ca^2+^ and Mg^2+^ ions are described by the factor FP-2 (Figure 9b). This process mainly covers 60% of the surface area of the aquifer, where the largest agricultural production in the La Yarada-Los Palos district is irrigated.

#### 4.3.3. Wastewater Mixture (NO_3_^−^)

The influence of wastewater delimited and described in factor FP-3 and FT-3 (Figure 9c), in areas adjacent to the Tacna wastewater treatment plants (Magollo and Copare) and the Tacna Municipal Forest, irrigated with wastewater where NO_3_^−^ infiltrations occur.

#### 4.3.4. Reducing Conditions

Favored by the long run of water in the subsoil and the depth at which it is located, depleting of oxygen the aquifer groundwater, described, and delimited by the factor FO-1 (Figure 10a).

#### 4.3.5. Contributions from (B, Sr)

Delimited and described by the factor FO-2 (Figure 10b), coming from the infiltration of the Caplina river contaminated by volcanic activity into the headwaters of its basin.

#### 4.3.6. Conservative Mixtures and Dissolution (As, F)

Delimited and described in the factor FO-3 (Figure 10c), show a local meteoric origin due to the low flow velocity and low recharge in the area, predisposing the water–rock contact, generating dissolution of feldspars and biotite in contact with solutions from the conservative mixture between seawater and freshwater with long residence times.

### 4.4. Quality of Drinking Water

The water quality index (WQI) was used [40], considering the standard limits proposed by the Peruvian Ministry of Health (MINSA). Concentrations of HCO_3_^−^, K^+^, and Sr were not included in the calculation because they do not have a limit prescribed by MINSA. The parameters TDS, Cl^−^, SO_4_^2−^, NO_3_^−^, B, As, and F were assigned the value 5, which is the one with the greatest weight due to its greater impact on water quality [64]. The pH has a relatively minor role in the quality of the water, receiving a minimum weight of 1 [65]. The remaining parameters were assigned a weight that varies between 1 and 5. The assigned weight, the calculated relative weight, and the limit prescribed by the MINSA for the quality parameters are presented in Table 4.

According to the WQI values, the quality of water for human consumption can be classified into different classes, such as excellent (<50), good (50–100), poor (100–200), very poor (200–300), and inadequate (>300) [37]. The spatial distribution of these indices is shown in Figure 13.

The values obtained ranged between excellent and very poor classes. A total of 70% of the pumping wells studied have “good” quality water, and to the SE, “excellent” quality wells; being a total of 85% that provides safe water for direct consumption. Parallel to the coastline, areas were determined that present “poor” and “very poor” quality for consumption and should not be used for drinking without some prior treatment otherwise there could be various health problems. These areas are not suitable for direct consumption, resulting from the combination of natural processes (interaction of water with the geological environment) and anthropogenic (infiltration of agrochemicals, wastewater, and overexploitation of wells that produces intrusion of seawater). 

### 4.5. Water Quality for Irrigation Purposes

The samples analyzed from irrigation wells were evaluated in the Wilcox diagram based on sodium content (%Na) and are distributed in Figure 14, also considering their cluster grouping.

The wells belonging to clusters 1, 2, and partially from cluster 3 belonged to the excellent, good, admissible, and ordinary category for irrigation. These can be used normally, in the case of ordinary, they can be used for irrigation in the absence of alternative water sources. However, half of the wells in cluster 3, located near the coastline, were unsuitable for irrigation purposes, registering an average of 3700 μS cm^−1^. The degradation of the quality of irrigation water was due to the increase of sodium and salinity, this was also observed in other parts of the world [66,67].

## 5. Conclusions

The contrast of areas with the highest values of the three factorial analysis approaches, it was possible to delimit the area of maximum influence by the hydrogeochemical processes that govern the degradation of water quality in La Yarada aquifer: (i) salinization—intrusion marine, (ii) fertilizer leaching and dissolution (Ca^2+^, Mg^2+^), (iii) wastewater mixture (NO_3_^−^), (iv) reducing conditions (Fe, Mn, Al), (v) contributions of (B, Sr), (vi) conservative mixtures and dissolution (As, F).Integration of regional geology with the few recharge areas in Caplina basin, show a preferential direction by the structural lineaments (NE-SO), forming different facies, grouping wells according to their similarity: (i) C1 low mineralized waters, with minimum influence of hydrogeochemical processes, (ii) C2 waters affected by marine intrusion with low recharges, (iii) C3 waters affected by marine intrusion and other mixtures with high recharges.It was validated with the water quality indices (WQI), that the areas near the coast are “bad” to “very bad” for human consumption. Therefore, they should not be used for drinking without any prior treatment otherwise health problems could arise.It was validated with the Wilcox diagram that the areas near the coast are unsuitable for irrigation, and this is due to the increase in sodium and salinity.

## Figures and Tables

**Figure 1 ijerph-19-02815-f001:**
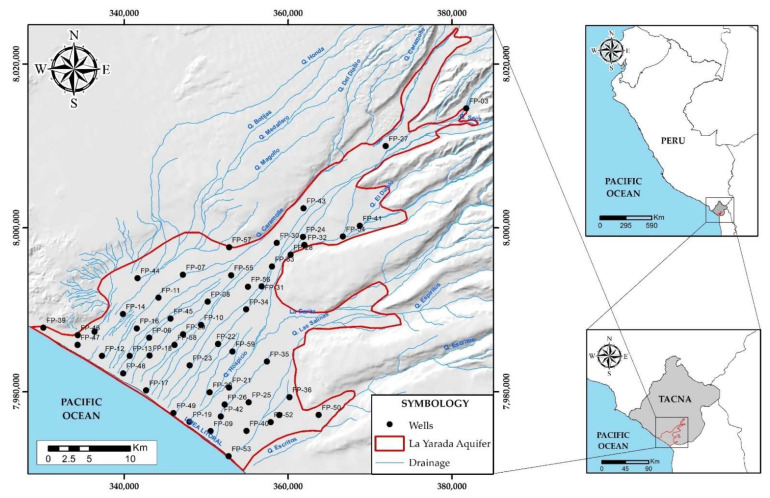
Location of the aquifer of La Yarada.

**Figure 2 ijerph-19-02815-f002:**
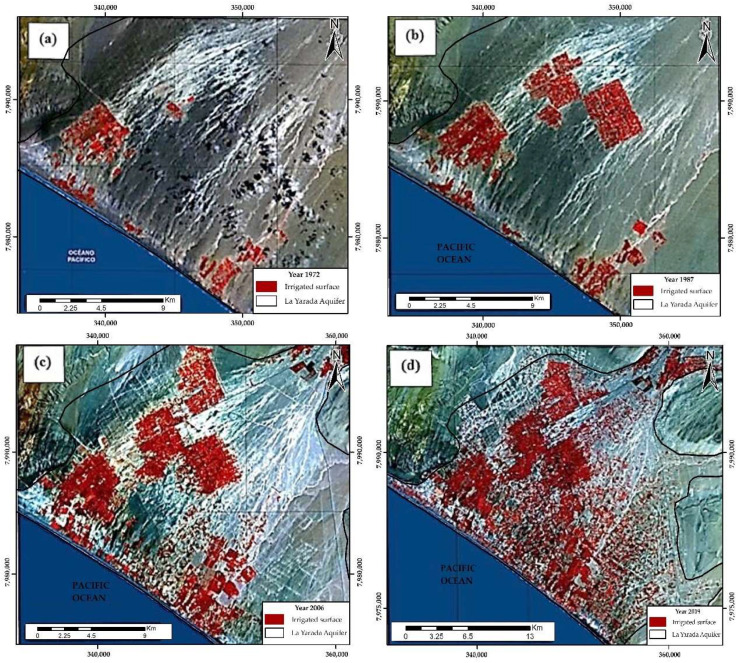
Evolution of irrigated area during (**a**) 1972, (**b**) 1987, (**c**) 2006 and (**d**) 2019, the sustained increase in the cultivated area is evidenced, and the corresponding increase in the exploited volume of the aquifer [21].

**Figure 3 ijerph-19-02815-f003:**
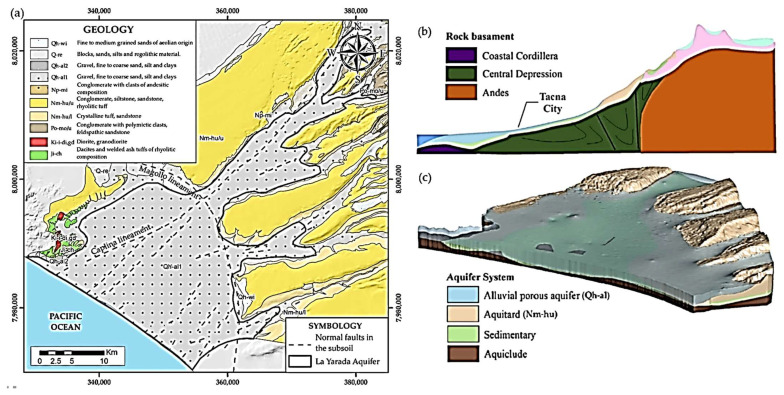
(**a**) Geological—structural map of La Yarada aquifer, (**b**) geomorfology Cross-section in Caplina Basin (**c**) 3D view of La Yarada aquifer [2].

**Figure 4 ijerph-19-02815-f004:**
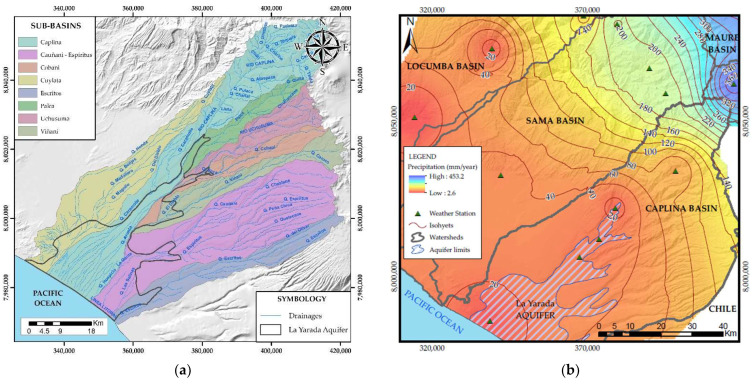
(**a**) Sub-basins of the Caplina basin. (**b**) Precipitation isohyets in the Caplina basin.

**Figure 5 ijerph-19-02815-f005:**
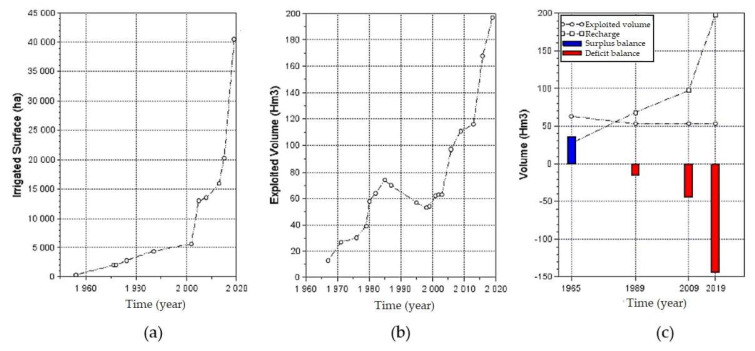
(**a**) Irrigated area (Ha). (**b**) Exploited volume (Hm^3^). (**c**) Water balance [21].

**Figure 6 ijerph-19-02815-f006:**
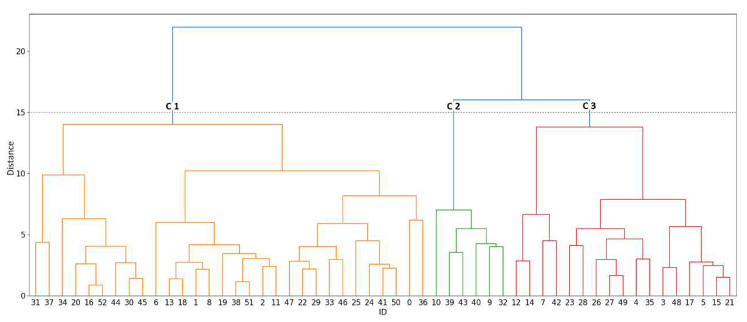
Cluster analysis dendrogram for 53 groundwater samples from the aquifer of La Yarada.

**Figure 7 ijerph-19-02815-f007:**
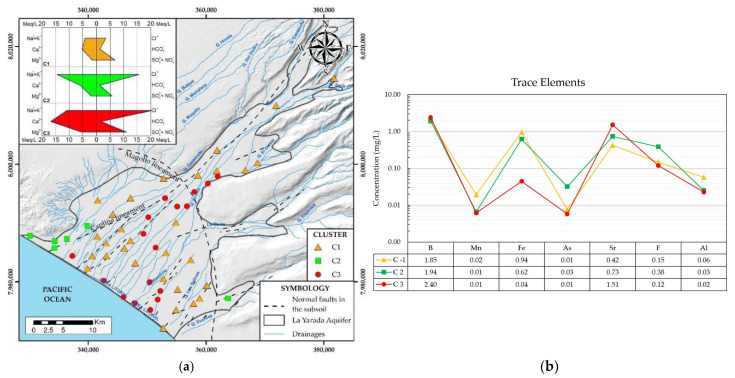
(**a**) Spatial distribution of the clusters. (**b**) Comparative diagram of trace elements between clusters.

**Figure 8 ijerph-19-02815-f008:**
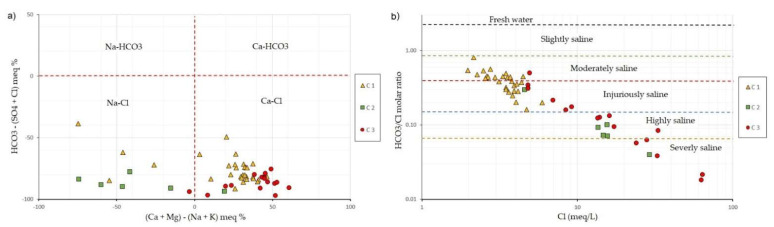
(**a**) Chadha diagram. (**b**) Simpson’s degrees of salinization.

**Figure 9 ijerph-19-02815-f009:**
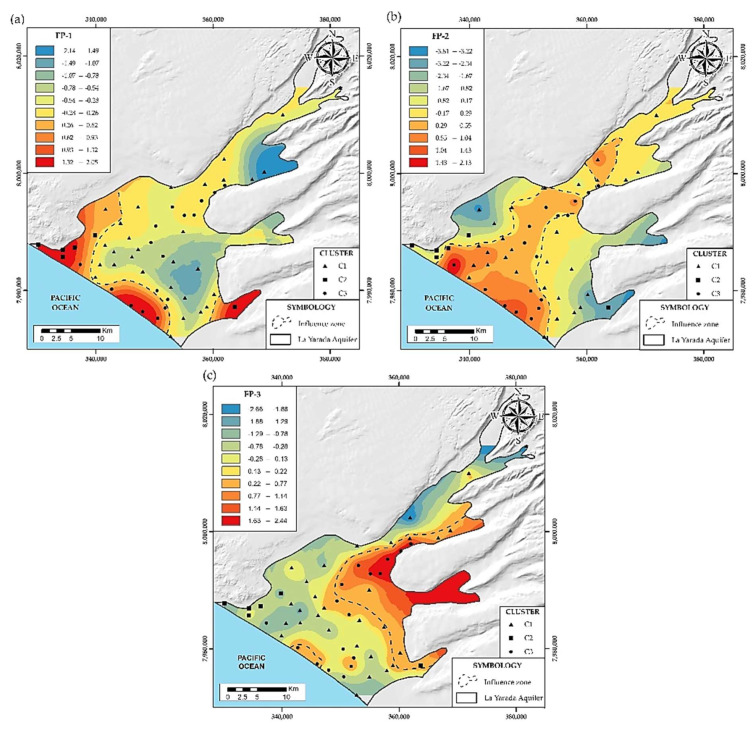
(**a**) Spatial distribution of factor FP-1. (**b**) Spatial distribution of factor FP-2. (**c**) Spatial distribution of factor FP-3.

**Figure 10 ijerph-19-02815-f010:**
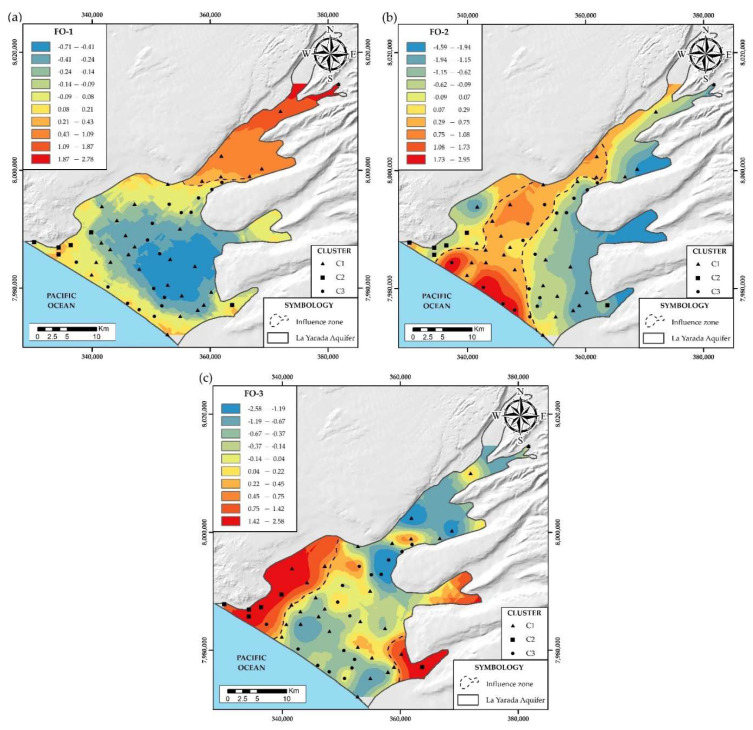
(**a**) Spatial distribution of factor FO-1. (**b**) Spatial distribution of factor FO-2. (**c**) Spatial distribution of factor FO-3.

**Figure 11 ijerph-19-02815-f011:**
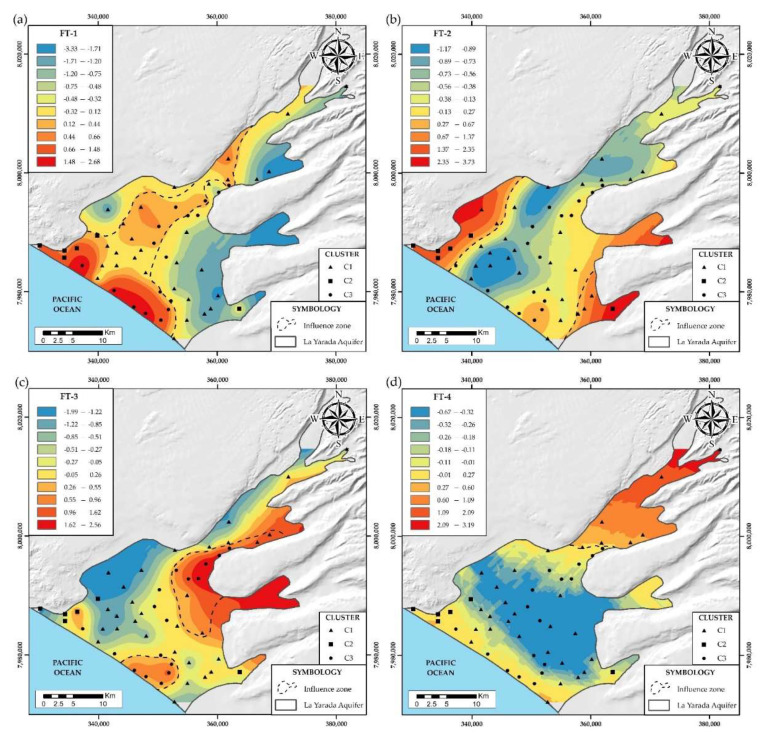
(**a**) Spatial distribution of factor FT-1. (**b**) Spatial distribution of factor FT-2. (**c**) Spatial distribution of factor FT-3. (**d**) Spatial distribution of factor FT-4.

**Figure 12 ijerph-19-02815-f012:**
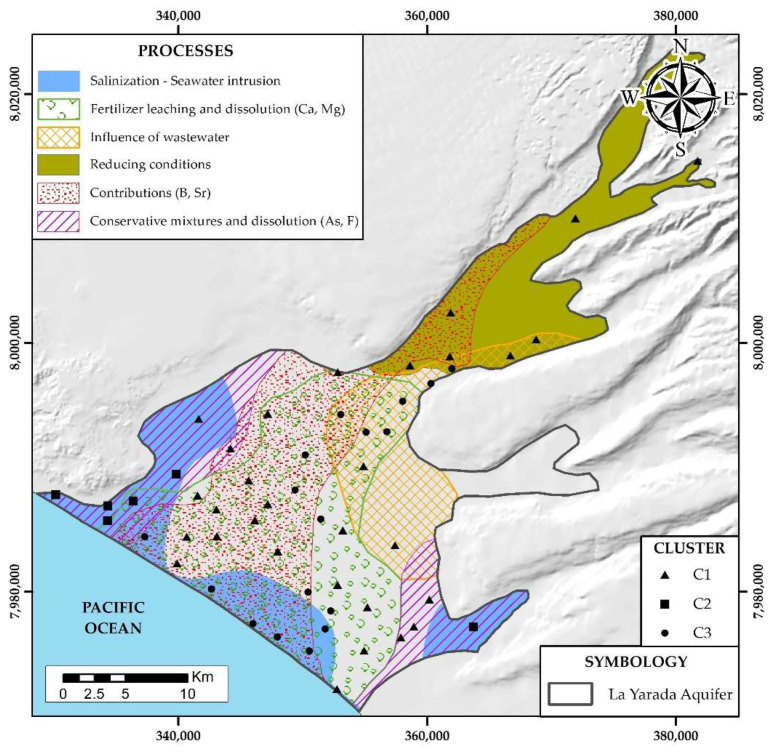
Hydrogeochemical processes that govern the water chemistry in the aquifer of La Yarada.

**Figure 13 ijerph-19-02815-f013:**
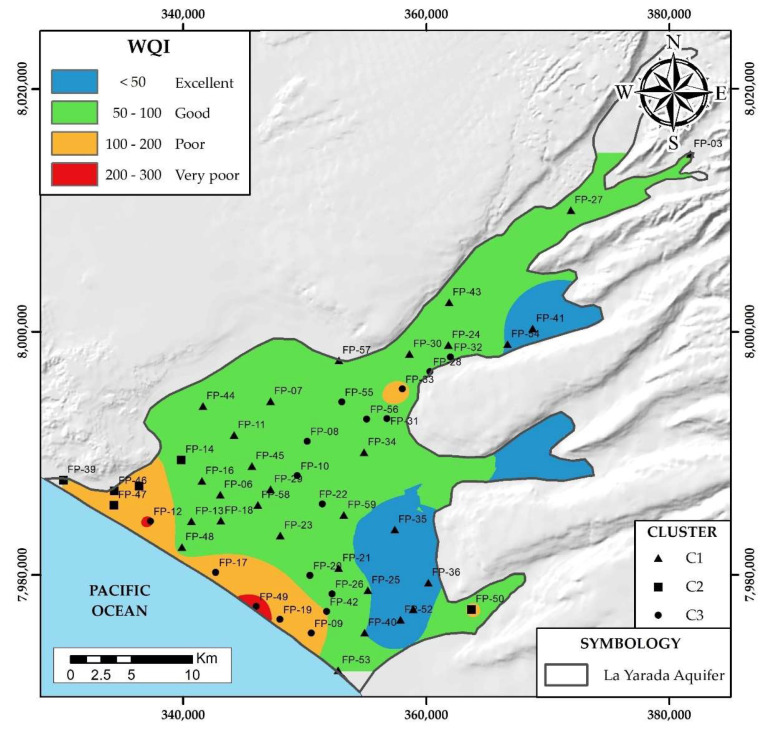
Spatial distribution of the quality of water for human consumption.

**Figure 14 ijerph-19-02815-f014:**
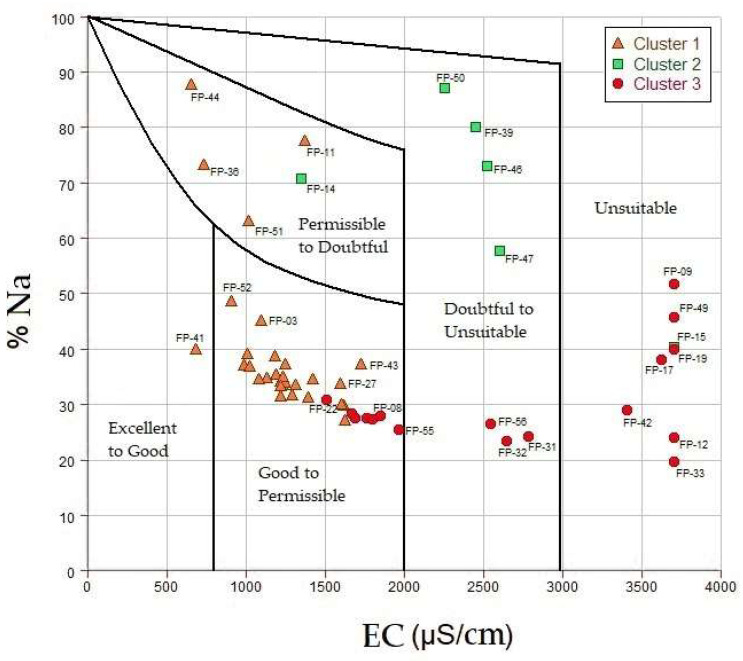
Classification of water for irrigation purposes.

**Table 1 ijerph-19-02815-t001:** Factors using main ions and physical properties.

FP	Factors		
	1	2	3
pH	−0.08	−0.68	−0.44
EC	**0.81**	0.54	0.23
TDS	**0.80**	0.54	0.24
Cl^−^	**0.76**	0.28	0.38
SO_4_^2−^	0.31	**0.74**	−0.02
NO_3_^−^	0.28	0.18	**0.94**
HCO_3_^−^	0.01	0.09	**0.69**
Ca^2+^	0.34	**0.86**	0.31
Mg^2+^	0.34	**0.93**	0.11
Na^+^	**0.98**	0.07	0.03
K^+^	**0.64**	0.40	−0.05
%Var	33	32	18
Acum	33	65	83

Words in bold shows the best values and a better reading and interpretation is achieved.

**Table 2 ijerph-19-02815-t002:** Factors using trace elements.

FO	Factors		
	1	2	3
B	0.02	**0.99**	0.04
Mn	**0.94**	0.06	−0.03
Fe	**0.87**	0.01	0.07
As	−0.24	−0.42	**0.51**
Sr	0.01	**0.65**	−0.25
F	0.08	−0.07	**0.99**
Al	**0.78**	0.03	−0.10
%Var	33	23	19
Acum	33	56	75

Words in bold shows the best values and a better reading and interpretation is achieved.

**Table 3 ijerph-19-02815-t003:** Factors using main ions, trace elements, physical properties, and stable isotopes.

FT	Factors			
	1	2	3	4
$\delta^{2}H$	0.04	**0.78**	−0.02	0.10
$\delta^{18}O$	−0.07	**0.83**	0.17	0.18
pH	−0.35	0.43	−0.69	−0.15
EC	**0.88**	0.15	0.43	0.09
TDS	**0.87**	0.16	0.44	0.10
Cl^−^	**0.63**	0.37	**0.58**	0.08
SO_4_^2^^−^	**0.83**	−0.47	0.01	0.01
NO_3_^−^	0.25	0.08	**0.78**	−0.34
HCO_3_^−^	0.05	−0.02	**0.51**	−0.38
Ca^2+^	**0.71**	−0.36	**0.56**	0.05
Mg^2+^	**0.79**	−0.41	0.37	0.05
Na^+^	**0.77**	0.52	0.11	0.10
K^+^	**0.78**	0.14	0.03	−0.05
B	**0.86**	−0.30	−0.24	−0.01
Mn	0.06	0.03	−0.03	**0.87**
Fe	0.02	0.10	−0.10	**0.86**
As	−0.30	**0.55**	−0.06	−0.33
Sr	**0.82**	−0.26	0.40	0.05
F	0.05	**0.69**	−0.40	−0.05
Al	0.04	−0.02	−0.07	**0.80**
%Var	33	17	15	13
Acum	33	50	65	78

Words in bold shows the best values and a better reading and interpretation is achieved.

**Table 4 ijerph-19-02815-t004:** Assigned weights for quality parameters.

Water Quality Parameters	Permissible Limit MINSA	w_i_	W_i_
pH	6.5–8.5	1	0.02
TDS	1000	5	0.09
Cl^−^	250	5	0.09
SO_4_^2−^	250	5	0.09
NO_3_^−^	50	5	0.09
Ca^2+^	200	3	0.06
Mg^2+^	150	3	0.06
Na^+^	200	4	0.08
B	1.5	5	0.09
Mn	0.4	2	0.04
Fe	0.3	2	0.04
As	0.01	5	0.09
F	1.0	5	0.09
Al	0.2	3	0.06

## Data Availability

Not applicable, part of the data corresponds to reports from Peruvian public institutions that under agreement provided such information with academic purposes, but the majority of data are in Table 1.

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
