# Peer review of "Hydrogeochemical Characterization and Identification of Factors Influencing Groundwater Quality in Coastal Aquifers, Case: La Yarada, Tacna, Peru"

_ijerph, 2022, doi:10.3390/ijerph19052815_

Round 1
Reviewer 1 Report
Dear authors,
your paper “Hydrogeochemical Characterization and Identification of Factors Influencing Groundwater Quality in Coastal Aquifers, Case: La Yarada, Tacna, Peru” is an very interesting reading and has good English used. The article is written in accordance with the editorial requirements and very carefully. The results might be of interest to the research communities on hydrology and hydrogeology. I wish you all the best and success with further processing of the article. I do propose a few corrections:
Section 2.3
- there is no reference to Figure 4b in the text
Section 2.4
Line 106 – you made a reference to Fig. 5b
but
the reference to Fig. 5a is in Line 107 – correct please the sequence of these quotations
Section 4.2.
Line 246 – there is Table 1, but in this Section there is no reference to this Table
Section 4.2.1
Line 251 – there is the first reference to Table 1
Please correct location of Table 1
It seems that this article includes the second part of the research presented by the authors in the article: https://doi.org/10.3390/w13223161. Maybe it should be highlighted in this part.
Reviewer 2 Report
See the atatched file.

Reviewer 3 Report
The manuscript is very interesting and can serve as a very good template for researches in some other areas of the world that have similar issues. I suggest to the authors small corrections to make the work even better and more applicable in other areas as well.
- It is necessary to expand the Introduction section and connect this research with other research studies and similar areas around the world and why these are special, if they are
- In the Introduction it is mentioned excessive pumping and lowering of groundwater levels, while in the manuscript it is analyzed only annual recharge and annual pumping rate and water deficit as a result, but the groundwater levels changes was not mentioned. I suggest an additional analysis of changes in groundwater levels over the period from 1972 to nowadays.
- In the Conclusion chapter it is necessary to extend and emphasize the applicability of this method to other arid and semiarid areas, as well as possible limitations of the method if any.
- A more detailed description of the geological formations of Huaylillas and Moquegua mentioned in the manuscript is needed.
- Below are comments on the figures that has to be corrected:
- In Figure 2 the legend is illegible, the graphical representation of the scale is illegible and blurred. In Figure 2d a different scale than in 3a, 3b and 3c. The La Yarada aquifer should also be shown in the pictures. If it is not the original image, I suggest digitizing the polygons of irrigated areas and displays as vector data.
- Figure 3 (Geological-structural map) - structural elements (faults) are missing or rename the figure.
- Figure 4 - translate the legend completely into English
- Figure 5 - larger letters in the legend
- Figure 8a - larger letters
- Figure 8b - the graph should be in English - translate and enlarge the letters
- Figure 9 - it is necessary to increase the descriptions for better visibility of the image
- Figure 13 - enlarge the legend - weakly hanging
- In Figure 14 it is necessary to highlight the classification of irrigation water and translate it into English
